# MIXED-PRECISION INFERENCE QUANTIZATION: RADICALLY TOWARDS FASTER INFERENCE SPEED, LOWER STORAGE REQUIREMENT, AND LOWER LOSS

## ABSTRACT

Model quantization is important for compressing models and improving computing speed. However, current researchers think that the loss function value of the quantized model is usually higher than the full-precision model. This study provides a methodology for acquiring a mixed-precision quantization model with a lower loss without "fine-tuning" than the full-precision model. Using our algorithm in different models on different datasets, we gain lower loss quantized models than full-precision models.

## 1    INTRODUCTION

Neural network storage, inference, and training are computationally intensive due to the massive parameter sizes of neural networks. Therefore, developing a compression algorithm for machine learning models is necessary. Model quantization, based on the robustness of computational noise, is one of the most important compression techniques. The primary sources of noise are truncation and data type conversion errors. In the quantization process, the initial high-precision data type used for a model's parameters is replaced with a lower-precision data type. Both PyTorch and TensorFlow have quantization techniques that translate floats to integers. Various quantization techniques share the same theoretical foundation, which is the substitution of approximation data for the original data in the storage and inference processes. A lower-precision data format requires less memory, and using lower-precision data requires fewer computer resources and less time. In quantization, the precision loss in different quantization level conversions and data type conversions is the source of the noise.

Current works, on the other hand, raise the following issues:1. No study examines how to reduce the loss function value of a model using quantization technology. There is the myth that the quantized model's loss is higher than the full-precision model. 2. The background of some work is against current computation device requirements: current computation devices have to use the same two data types in one computation process, which means the layer's weight and the input of the layer have to be the same quantization level. 3. No one has examined which types of models are stable in the quantization process and why.

The purpose of this paper is mainly to discuss the question of whether quantization technology always leads to the model's loss function increasing and how to gain a better performance quantized model by quantization method. In current papers, the main target of the current algorithm is to gain a quantized model whose loss function value is not much higher than a full-precision model. However, we want to give an algorithm that can find the quantized model that is better than the full precision model, i.e., the quantized model's loss function value is lower than the full precision model, based on the current computation device's requirements.

This research provides a basic analysis of the computational noise robustness of neural networks. Furthermore, we present a method for acquiring a quantized model with a lower loss than the model with full precision by using the $floor$ and $ceiling$ functions in different layers, with a focus on layerwise post-training static model quantization.

As an added benefit in algorithm analysis, we give the theoretical result to answer the question that which types of models are stable in the quantization process and why when the noise introduced by quantization process can be covered by the neighborhood concept.

## 2 RELATED WORK

Model compression methods include pruning methodsHan et al. (2015); Li et al. (2016); Mao et al. (2017) , knowledge distillationHinton et al. (2015), weight sharingUllrich et al. (2017) and quantization methods. From the perspective of the precision layout, post-training quantization methods can be mainly divided into channelwise Li et al. (2019); Qian et al. (2020), groupwise Dong et al. (2019b) and layerwise Dong et al. (2019a) methods. Layerwise mixed-precision layout schemes are more friendly to hardware. Parameters of the same precision are organized together, making full of a program's temporal and spatial locality. Some works give the relationship between the weight and input of layer's best quantization analysisSakr et al. (2017); Sakr & Shanbhag (2018). But in current computation architectures, the quantization level for weight and input should be the same. A common problem definition for quantizationDong et al. (2019a); Morgan et al. (1991); Courbariaux et al. (2015); Yao et al. (2020) is as follows Gholami et al. (2021).

**Problem 1** *The objective of quantization is to solve the following optimization problem:*

$$\min_{q \in \mathbf{Q}} \|q(w) - w\|^2$$

*where $q$ is the quantization scheme, $q(w)$ is the quantized model with quantization $q$, and $w$ represents the weights, i.e., parameters, in the neural network.*

Although problem 1 gives researchers a target to aim for when performing quantization, the current problem definition has two shortcomings: 1. The search space of all possible mixed-precision layout schemes is a discrete space that is exponentially large in the number of layers. There is no effective method to solve the corresponding search problem. 2. There is a gap between the problem target and the final task target. As we can see, no terms related to the final task target, such as the loss function or accuracy, appear in the current problem definition.

## 3 BACKGROUND ANALYSIS

### 3.1 MODEL COMPUTATION, NOISE GENERATION AND QUANTIZATION

Compressed models for the inference process are computed using different methods depending on the hardware, programming methods and deep learning framework. All of these methods introduce noise into the computing process. One reason for this noise problem is that although it is common practice to store and compute model parameters directly using different data types, only data of the same precision can support precision computations in a computer framework.

Therefore, before performing computations on nonuniform data, a computer will convert them into the same data type. Usually, a lower-precision data type in a standard computing environment will be converted into a higher-precision data type; this ensures that the results are correct but require more computational resources and time. However, to accelerate the computing speed, some works on artificial intelligence (AI) computations propose converting higher-precision data types into lower-precision data types based on the premise that AI models are not sensitive to compression noise. The commonly used quantization technology is converting data directly and using a lower-precision data type to map to a higher-precision data type linearly.

We use the following example to illustrate quantization method, which is presented in Yao et al. (2020). Suppose that there are two data objects $input_1$ and $input_2$ are to be subjected to a computing operation, such as multiplication. After the quantization process, we have $Q_1 = \mathrm{int}(\frac{input_1}{scale_1})$ and $Q_2 = \mathrm{int}(\frac{input_2}{scale_2})$, and we can write

$$Q_{output} = \mathrm{int}(\frac{input_1 * input_2}{scale_{output}}) \approx \mathrm{int}(Q_1 Q_2 \frac{scale_1 * scale_2}{scale_{output}})$$

$scale_{output}$, $scale_1$ and $scale_2$ are precalculated scale factors that depend on the distributions of $input_1$, $input_2$ and the output; $Q_i$ is stored as a lower-precision data type, such as an integer. All

$scale$ terms can be precalculated and established ahead of time. Then, throughout the whole inference process, only computations on the $Q_i$ values are needed, which are fast. In this method, the noise is introduced in the $\text{int}(\cdot)$ process. This basic idea gives rise to several variants, such as (non)uniform quantization and (non)symmetrical quantization.

When we focus on quantization strategy, i.e. $round$ function in quantization framework like Micronet, we can have at least three strategy: round up, i.e., $ceil$ function in python, round down, i.e., $floor$ function in python and rounding, i.e., $round$ function in python. usually, rounding is the most common method to deal with quantization. But, in this paper, we will show that how to mixed use round up/round down to gain a mixed precision quantized model which is better than full precision model.

## 3.2 NEURAL NETWORKS

In this paper, we mainly use the mathematical properties of extreme points to analyze quantization methods. This approach is universal to all cases, not only neural networks. However, there is a myth in the community that it is the neural network properties that guarantee the success of quantization methodsWang et al. (2019); Morgan et al. (1991); Demidovskij & Smirnov (2020). To show that the properties of the extreme points, not the properties of the neural network, are what determine the ability to quantize, i.e. the ability to handle noise, we must first define what a neural network is.

The traditional definition of a neural network Denilson & Barbosa (2016) as a human brain simulation is ambiguous; it is not a rigorous mathematical concept and cannot offer any analyzable information. The traditional descriptions of neural networks Denilson & Barbosa (2016) focus on the inner products of the network weights and inputs, the activation functions and directed acyclic graphs. However, with the development of deep learning, although most neural networks still consist of weighted connections and activation layers, many neural networks no longer obey these rules, such as the network architectures for position embedding and layer norm operations in Transformers. Moreover, current deep learning frameworks, such as PyTorch and TensorFlow, offer application programming interfaces (APIs) to implement any function in a layer. Therefore, we propose that the definition of a neural network adheres to the engineering concept indicated by the definition 1 rather than a precise mathematical definition; that is, a neural network is a way for implementing a function.

**Definition 1** *The neural network is the function which is implemented in composite function form.*

A neural network can be described in the following Eq. 1 form.

$$model(x) = h_1(h_{2,1}(h_{3,1}(...), ..., h_{3,k}, w_{2,1}), h_{2,2}(h_{3,k+1}(...), ..., w_3), ..., w_{2,2}), ..., w_1) \quad (1)$$

where $h_{i,j}, i \in [2, ..., n]$, are the $(n - i + 1)$th layers in the neural network; $w_{i,j}$ is the parameter in $h_{i,j}(\cdot)$.

Definition 1 means that a neural network, without training, can be any function. With definition 1, a neural network is no longer a mathematical concept, but this idea is widely used in practice Roesch et al. (2019). We can see from definition 1 that the requirement that a neural network is in composite function form is the only mathematical property of a neural network that can be used for analysis.

In practice, the loss function is one method to evaluate a neural network. A lower loss on a dataset means a better performance neural network. For example, the training process optimises the model's loss, i.e., following Eq. 2.

$$\min_w f(w) = \mathbf{E}_{sample}\ell(w, sample) = \frac{1}{m} \sum_{(x_i, y_i) \in \mathbb{D}} \ell(w, x_i, y_i) \quad (2)$$

where $f(\cdot)$ is the loss for model on a dataset, $w$ represents the model parameters, $\mathbb{D}$ is the dataset, $m$ is the size of the dataset, $\ell(\cdot)$ is the loss function for a sample and $(x_i, y_i)$ represents a sample in the dataset and its label.

In this paper, we mainly use the sequential neural network to describe the conclusion for the sequential neural network is easily described, and the whole conclusion is non-related to the structure of the neural network. For a sequential $n$-layer neural network, $\ell(\cdot)$ can be described in the following Eq.3 form.

$$\ell(w, x_i, y_i) = L(model_n(x_i, w), y_i)$$
$$model_n = h_1(h_2(h_3(h_4(\cdots h_n(h_{n+1}, w_n) \cdots, w_4), w_3), w_2), w_1) \quad (3)$$

where $L(\cdot)$ is the loss function, such as the cross-entropy function; $h_i$, $i \in [1, ..., n]$, is the $(n-i+1)$th layer in the neural network; $w = (w_n^T, w_{n-1}^T, \cdots, w_1^T)^T$, $w_i$ is the parameter in $h_i(\cdot)$; and for a unified format, $h_{n+1}$ stands for the sample $x$.

# 4 ALGORITHM AND BASIC ANALYSIS

## 4.1 START POINT

### 4.1.1 ANALYSIS BASE

Quantization methods for inference are complex. Different algorithms use different assumption to solve the problem. Most of them pay much attention to the noise on parameters in NNDong et al. (2019a); Yao et al. (2020); Gholami et al. (2021); Nagel et al. (2020). However, in addition to the noise added to the parameters directly, noise is also introduced between different layers in the inference process because different quantization levels or data types of different precisions are used in different layers.

After quantization, the quantized loss for a sample, i.e. $\bar{\ell}(\cdot)$, in the inference process is as follows.

$$\bar{\ell}(w, x_i, y_i) = L(h_1(h_2(\cdots h_n(h_{n+1} + \epsilon_n, w_n + \delta_n) + \epsilon_{n-1} \cdots, w_2 + \delta_2) + \epsilon_1, w_1 + \delta_1), y_i)$$

where $\delta_i$, $i \in 1, \cdots, n$, and $\epsilon_i$, $i \in [1, ..., n]$, are the minor errors that are introduced in model parameter quantization and in data type conversion in the mixed-precision layout scheme, respectively.

Thus, we obtain the following expression based on the basic total differential calculation.

$$\bar{\ell}(w, x_i, y_i) - \ell(w, x_i, y_i) = \sum_{i=1}^{n} \frac{\partial \ell}{\partial h_{i+1}} \cdot \epsilon_i + \frac{\partial \ell}{\partial w_i} \cdot \delta_i \tag{4}$$

where $\cdot$ is inner product and $*$ is the scalar product in following parts. For the loss on whole dataset, we can gain

$$\min_{\epsilon \in E} \bar{f}(w) - f(w) = \frac{1}{m} \sum_{(x_j, y_j) \in \mathbb{D}} \sum_{i=1}^{n} \frac{\partial \ell}{\partial h_{i+1}} \cdot \epsilon_i + \frac{\partial \ell}{\partial w_i} \cdot \delta_i = \frac{1}{m} \sum_{i=1}^{n} \sum_{(x_j, y_j) \in \mathbb{D}} \frac{\partial \ell}{\partial h_{i+1}} \cdot \epsilon_i \tag{5}$$

where $\bar{f}(w) = \frac{1}{m} \sum \bar{\ell}(\cdot)$. The reason for second equation in Eq. 5 is for a well-trained model, the expectation of $\ell(\cdot)$'s gradient for parameters is zero, i.e., for the $\sum_{(x_j, y_j) \in \mathbb{D}} \frac{\partial \ell}{\partial w}$ components, $\frac{\partial \ell}{\partial w_i} = 0$.

### 4.1.2 TARGET AND ALGORITHM GUARANTEE

The key is to choose the appropriate $\epsilon$ vector to gain a lower loss model. When the loss of the inner product, i.e., $\sum_{(x_i, y_i) \in \mathbb{D}} \frac{\partial \ell}{\partial h_{i+1}} \cdot \epsilon$, is negative, the loss for the quantized model, i.e., $\bar{f}$, is lower than for the full precision model. An appropriate $\epsilon$ to produce a negative $\sum_{(x_i, y_i) \in \mathbb{D}} \frac{\partial \ell}{\partial h_{i+1}} \cdot \epsilon$ is our algorithm target.

A frequently asked question is why $\sum_{(x_j, y_j) \in \mathbb{D}} \frac{\partial \ell}{\partial w}$ is zero but $\sum_{(x_i, y_i) \in \mathbb{D}} \frac{\partial \ell}{\partial h_{i+1}}$ is non-zero. The optimization algorithm is to optimize $w$ in the training process. Thus, $\sum_{(x_i, y_i) \in \mathbb{D}} \frac{\partial \ell}{\partial h_{i+1}}$ is random in the final model except for the layers with bias terms like the batch norm layer. The bias term will absorb the gradient and train them in the optimization process. What is more, in the model, which mainly consists of identity mapping, $\sum_{(x_i, y_i) \in \mathbb{D}} \frac{\partial \ell}{\partial h_{i+1}}$ is close to zero vector, and we will show this in the next chapter.

Our problem setting for quantization is different from previous works like HAWQDong et al. (2019a); Yao et al. (2020); Dong et al. (2019b); Nagel et al. (2020) because these methods do not take the error in the layer's input into consideration, which prevents their work and analysis in the mixed-precision computing area. As a result, these works can only be used to store a compressed neural network on a disk. When the compressed model is stored in memory for inference, these compressed models have to be recovered into the full precision model.

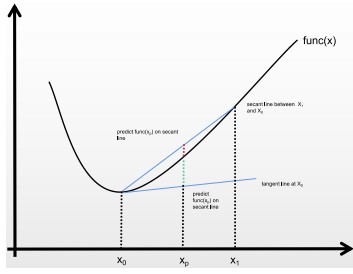

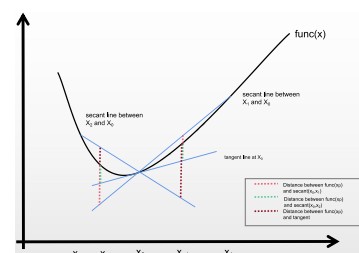

Figure 1: When the predicted point($x_p$) is out of the neighborhood range but not pretty far from $x_0$, secant lines between the $x_0$ and $x_1$ perform significantly better than tangent lines. The choice of $x_1$ is the maximum quantization noise in practice.

Figure 2: Different directions have different secant line in prediction process.

## 4.2 THE MAP FROM MATHEMATICAL ANALYSIS TO REAL ENGINEERING

In the above analysis, the whole process is under the condition that $\epsilon$ vector is small enough, which can be used in the total differential method. However, in practice, the scope of $\epsilon$ may be within [-0.1,0.1], which would escape the concept of neighborhood. What is more, mapping $\epsilon$ vector into round operation should be fully discussed. This part will show how to deal with the above gap between analysis and engineering.

### 4.2.1 ROUND FUNCTION CHOICE

We use the convenient language of probability theory to describe $\frac{\partial \ell}{\partial h_{i+1}} \cdot \epsilon$ for $\epsilon$ is a stochastic vector naturally. We set $\epsilon = [e_1, e_2, .., e_k]$ and $e_i$ is i.i.d. random variable. We also set that $\frac{\partial \ell}{\partial h_{i+1}} = [p_1, p_2, ..., p_k]$ and $p_i$ is i.i.d. random variable [1]. $e$ and $p$ are independence to each other.

Then, we have $\frac{\partial \ell}{\partial h_{i+1}} \cdot \epsilon = \sum_{i=1}^{k} e_i * p_i = kep$ and following Eq. 6.

$$\mathbf{E}\frac{\partial \ell}{\partial h_{i+1}} \cdot \epsilon = \mathbf{E}\sum_{i=1}^{k} e_i * p_i = \mathbf{E}kep = k\mathbf{E}e\mathbf{E}p \qquad (6)$$

For a trained model, the $\mathbf{E}p$ can be computed as $\mathbf{E}p = \frac{1}{k} * \frac{\partial \ell}{\partial h_{i+1}} \cdot \vec{1}$. Then to gain a negative $\mathbf{E}\frac{\partial \ell}{\partial h_{i+1}} \cdot \epsilon$, the $\mathbf{E}e$ should be different signs with $\mathbf{E}p$.

To gain the suitable $\epsilon$ vector, we use the different round functions to ensure the sign of $\mathbf{E}e$. The roundup function, i.e., the $ceil$ function in python, will produce an error vector whose all elements are positive. The round down function, i.e., the $floor$ function, will produce an error vector whose all elements are negative. Thus, we are sure that the $\mathbf{E}e$ is positive and negative by round methods. Although the parameters in layers have strong noise robustness, we still try to add less noise to them. Thus, in the parameters quantization process, we use the rounding method, i.e., the $round$ function in python, to quantize parameters for the rounding method exerts less noise on original data.

### 4.2.2 REPLACE GRADIENT WITH SECANT LINE SLOPE

Although the elements in the $\epsilon$ vector are not small enough to use the total differential directly, the elements in the $\epsilon$ vector are still small. For example, when using INT8 to quantize the res14 model without identity mapping, the element in the $\epsilon$ vector is less than 0.01. The above fact shows that $\|\epsilon_i\| * \|\epsilon_i\|$ is small, which has a tiny influence on the final loss function. Thus, we can use the slope of the secant line to replace the gradient in the total differential, which is shown in figure 1.

---

[1] We also can treat $p_i$ as the random variable with different distributions or directly use $\mathbf{E}\frac{\partial \ell}{\partial h_{i+1}}$ vector in following analyses. The conclusions are the same or close with current analysis.

---

**Algorithm 1:** Radical Mixed-Precision Inference Layout Scheme

---

**Input:** Neural network $M$, quantization levels $[q_1, q_2, ..., q_n]$, $error_{min}$, $error_{max}$, $\mu$, calibration dataset $D$

**Output:** Quantized neural network $\bar{M}$

Arrange $Q = [q_1, q_2, ..., q_n]$ in ascending order $Q = [q_{i_1}, q_{i_2}, ..., q_{i_n}]$ based on the the size of parameters under $q_i$;//For example Q=[INT8,INT4,INT16] into Q=[INT4,INT8,INT16]

**for** $q_i$ *in* $Q$ **do**

    **for** $Layer_i$ *in* $M$ **do**

        **if** $Layer_i$ *is quantized* **then**

            | continue

        **end**

        Compute the error $\Delta$ of $h_{i+1}$ under $q_i$ quantization level on $D$

        **if** $\Delta < error_{min}$ **then**

            | quantize $Layer_i$'s parameters and input by rounding method on $q_i$ quantization level.

        **end**

        **if** $\Delta > error_{max}$ **then**

            | continue

         **end**

        Compute $secant^+(h_i, \Delta)$ and $secant^-(h_i, \Delta)$

        **if** $\|secant^{\pm}(\cdot)\| < \mu$ **then**

            | continue

        **end**

        $Choice = \max( \| \max(secant^-(h_i, \Delta), 0)\|, \| \min(secant^+(h_i, \Delta)0)\|,)$

        **if** $Choice == secant^-(h_i, \Delta)$ **then**

            | quantize $Layer_i$'s input by round down method on $q_i$ quantization level.

        **end**

        **if** $Choice == secant^+(h_i, \Delta)$ **then**

            | quantize $Layer_i$'s input by round up method on $q_i$ quantization level.

        **end**

        **if** $Choice$ *!= 0* **then**

            | quantize $Layer_i$'s parameters by rounding method on $q_i$ quantization level.

        **end**

    **end**

**end**

**return** $\bar{M}$

---

Because in different direction, the secant line is different, which is shown in figure 2, so we have to define the following $secant^+(h_i, \Delta), \Delta \in \mathbb{R}^1+$ and $secant^-(h_i, \Delta), \Delta \in \mathbb{R}^1+$. $\Delta$ is the maximum error which is introduced by quantization. For example, the *scale* parameter in Section 3.1's example is the max error introduced by quantization.

$$\bar{\ell}_j^{\bar{\pm}}(w, x_i, y_i) = L(h_1(h_2(\cdots h_j(h_{j+1}(\cdots) \pm \Delta * \vec{1}, w_n) \cdots, w_2), w_1), y_i), \Delta > 0$$

$$\bar{f}_j^{\pm}(w) = \frac{1}{m} \sum_{x_i, y_i \in Dataset} \bar{\ell}_j^{\pm}(\cdot), secant(h_i, \Delta)^{\pm} = \frac{f^{\pm} - f}{\pm \Delta}$$

In the algorithm, we will use $scant^{\pm}(\cdot)$ to replace $\frac{\partial \ell}{\partial h_{i+1}}^{\pm} \cdot \vec{1}$. We use this definition because 1. Compared to computing by the definition of secant, the *secant* function is easy to be computed. 2. If slope of secant line is $[sec_1, sec_2 ..., sec_k]$, $secant(h_i, \Delta)^{\pm} = \mathbf{E}sec$.

Although we know the element in $\epsilon$ vector is less than 0.01 empirically, we still have to set a mechanism in real algorithm design to keep the analysis map into algorithm practice. Thus, we have to set a value $error_{max}$, which $\epsilon$ is small enough for the final loss function. When $\Delta > error_{max}$, we can choose more bits quantization level or full precision in this layer.

When $\epsilon$ is close to zero, i.e., we use more bits quantization level. For $\|\frac{\partial \ell}{\partial h_{i+1}} \cdot \epsilon_i\| \leq \|\frac{\partial \ell}{\partial h_{i+1}}\|\|\epsilon_i\|$, the performance loss or improvement is small on this layer. So, we directly quantize these parameters

and layer's input with this quantization level to reduce computation resources. In algorithm design, we can use $error_{min}$ to control this case.

### 4.2.3 THE PROBABILITY OF GETTING A BETTER MODEL

To show the probability of getting positive $\frac{\partial \ell}{\partial h_{i+1}} \cdot \epsilon$, we use chebyshev's theorem, we have following Eq. 7.

$$P(\frac{\partial \ell}{\partial h_{i+1}}\epsilon \geq 0) < P(\|\frac{\partial \ell}{\partial h_{i+1}}\epsilon - \mathbf{E}kep\| \geq \|\mathbf{E}kep\|) \leq \frac{Var(kep)}{\|\mathbf{E}kep\|^2} = \frac{Var(e)Var(p)}{\|\mathbf{E}e\mathbf{E}p\|^2} + \frac{Var(e)}{\|\mathbf{E}e\|^2} + \frac{Var(p)}{\|\mathbf{E}p\|^2}$$

(7)

Based on Eq.7, we know that to gain a better model performance, for the layer whose $\|\frac{\partial \ell}{\partial h_{i+1}} \cdot \vec{1}\|$ is large and $Var(p)$ is small, we can use high quantization level to gain a model which is better than full precision model with high probability. To guarantee the success probability is high, we can set a algorithm parameter $\mu$. Algorithm quantize $Layer_i$ only when $\|\mathbf{E}p\| > \mu$.

### 4.3 ALGORITHM DESCRIPTION

Based on the above map between analysis and engineering, we can get algorithm 1. Algorithm 1 is a radical probability algorithm. In algorithm 1, we use a high quantization level as a priority to gain a small quantized model. Under the appropriate $\mu$ setting, algorithm 1 would give a better model with a high probability.

## 5 THE LIMITATION OF ALGORITHM 1 AND MODEL ROBUSTNESS

Although algorithm 1 provides a better model, yet in experiments, we find that ResNet50 / ResNet101 are hard to gain a significant improvements, which makes us think our algorithm have the limitations. To show this limitations is rooted in the model properties, we will prove a stronger conclusion in this section.

The neural network is under the description of the probably approximately correct (PAC) learning frameworkDenilson & Barbosa (2016). A neural network hypothesis class $\mathscr{H}$ consists of the neural networks which share the same structure. The learning algorithms, $\mathscr{A}$, are SGD and SGD's variants for the neural network hypothesis class. Identity mapping is when the input to some layer is passed directly or as a shortcut to some other layer. The neural networks, which mainly consist of identity mappings, like ResNet or DenseNet, succeed in the CV area. Then, we can gain the following propositions.

**Proposition 1** *There is a set of function $\mathscr{G}$. For any random variable vector $x$ and any random variable vector $y$, $\exists g \in \mathscr{G}$ which satisfies $\mathbf{E}g(x) \cdot \mathbf{E}y \leq 0$ and $g(x)$ belongs to $\vec{0}$'s neighborhood.*

*For a well-trained neural network $model_n^* \in \mathscr{H}_n$ by learning algorithm $\mathscr{A}$, there exists a $model_{n+1} \in \mathscr{H}_{n+1}$ which is slightly better than $model_n^*$. The difference between $\mathscr{H}_n$ and $\mathscr{H}_{n+1}$ is the $model_{n+1} \in \mathscr{H}_{n+1}$ have one more residual block than $model_{n+1} \in \mathscr{H}_n$ and the function in residual block is in $\mathscr{G}$.*

Brief proof: From the analysis in algorithm 1, we can find an appropriate $\mathbf{E}\epsilon$ that $\mathbf{E}\frac{\partial \ell}{\partial h_{i+1}} \cdot \epsilon \leq 0$. We can use $g \in \mathscr{G}$ to replace $\epsilon$. Then, proposition 1 is proved, which is also shown in figure 3.

The set, which consists of $Relu(Conv(\cdot))$, satisfies the requirements of $\mathscr{G}$. Proposition 1 tells us how to structure a deep residual network. Repeatedly using proposition 1 and retraining the new model would show that for the neural networks consisting of residual blocks like ResNet, the deeper, the better. It is shown in figure 4. Using proposition 1 in a different place, we can get different networks, like Resnet or DenseNet.

Based on the proposition 1's structure process, we can prove the following proposition 2.

**Proposition 2** *For a dataset's SOTA or close to SOTA residual network, all $\mathbf{E}\frac{\partial \ell}{\partial h_{i+1}}$ are close to zero.*

Brief proof: The SOTA model implies that adding new layers will not improve model performance, i.e., for well-trained $model_n^* \in \mathscr{H}_n$ and well-trained $model_{n+1}^* \in \mathscr{H}_{n+1}$, $\mathbf{E}_{sample}L(model_n^*, sample) - \mathbf{E}_{sample}L(model_{n+1}^*, sample) = 0$. So for any $i$ and any appropriate $\mathbf{E}\epsilon$, we have the following Eq 8.

$$\mathbf{E}\frac{\partial \ell}{\partial h_{i+1}} \cdot \epsilon = \mathbf{E}_{sample}L(model_n^*, sample) - \mathbf{E}_{sample}L(model_{n+1}, sample)$$

$$< \mathbf{E}_{sample}L(model_n^*, sample) - \mathbf{E}_{sample}L(model_{n+1}^*, sample) = 0 \qquad (8)$$

Because $i$ and $\mathbf{E}\epsilon$ can be chosen at random, we can tell that $\mathbf{E}\frac{\partial \ell}{\partial h_{i+1}}$ is zero or very close to zero.

Proposition 2 shows one of the residual network's SOTA criterion. Then, we can prove the following theorem 1.

**Theorem 1** *When quantization noise is under the concept of neighborhood, SOTA or near to SOTA residual networks in a dataset exhibit high noise robustness.*

Brief proof: Based on Eq.5 and proposition 2, we know

$$\|\bar{f}(w) - f(w)\| = \|\sum_{i=1}^{n} \mathbf{E}\frac{\partial \ell}{\partial h_{i+1}} \cdot \epsilon_i\| \leq \sum_{i=1}^{n} \|\mathbf{E}\frac{\partial \ell}{\partial h_{i+1}}\|\|\mathbf{E}\epsilon_i\| = 0 \qquad (9)$$

which means the noise would have higher-order infinitesimal, i.e., $o(noise)$, influence. Thus, SOTA or near to SOTA residual networks in a dataset exhibit high noise robustness

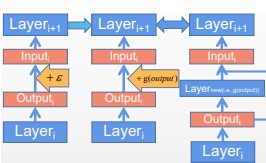 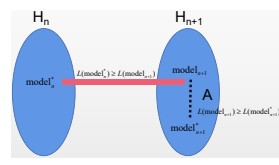

Figure 3: Proof of proposition 1

Figure 4: The deeper the neural network is, the better.

Theorem 1 shows that the robustness is stronger with increase of number of layers and identity mapping when quantization noise is under the concept of neighborhood. Based on Theorem 1, we can gain followint corollary.

**Corollary 1** *When the quantization noise is small, algorithm 1 cannot improve SOTA or near SOTA model's performance too much.*

## 6 EXPERIMENT

In this section, we evaluate the performance of algorithm 1. Our objective is to show that the quantized model gained by algorithm 1 is better than the full precision model without "fine-tuning" technology.

### 6.1 EXPERIMENT SETTING

#### 6.1.1 DATASET AND MODEL

We make use of datasets from MNIST, CIFAR 10, CIFAR 100, and ImageNet-100. The calibration and training datasets are separated from the training dataset. The calibration dataset's size is also the same as the test dataset's.

We employ a DNN as a benchmark in the MNIST dataset that is in accordance with the workSakr et al. (2017). ReLU layer is between each layer in the model's 784-512-256-128-64-10 design. We apply ResNet8/14 and VGG11/13 to the CIFAR 10 dataset. We employ VGG13, ResNet34, and mobilnet in the CIFAR 100 dataset. The mobilenet dataset for ImageNet-100 is used. We remove the identity mapping structure from the Resnet model in our experimental models to magnify the outcomes of the tests.

| model_dataset | full model's loss | quantized model | quantization level |
|---|---|---|---|
| vgg13_cifar10 | 0.0091 | **0.0086** | INT8&FLOAT |
| vgg11_cifar10 | 0.0019 | **0.0017** | INT8&FLOAT |
| res8_cifar10 | 0.3896 | **0.3782** | INT8&FLOAT |
| res14_cifar10 | 0.3634 | **0.3576** | INT8&FLOAT |
| CNN_mnist | 0.0792 | **0.0774** | INT8&INT4&FLOAT |
| CNN_mnist | 0.0792 | **0.0786** | INT8&FLOAT |
| CNN_mnist | **0.0792** | 0.0813 | INT8&INT4 |
| vgg13_cifar100 | 1.2726 | **1.2503** | INT8&FLOAT |
| vgg13_cifar100 | 1.2726 | **1.2384** | INT8&INT4&FLOAT |
| vgg13_cifar100 | **1.2726** | 2.4891 | INT8&INT4&int2&FLOAT |
| mobilenet_cifar100 | 1.5653 | **1.5631** | INT8&FLOAT |
| mobilenet_cifar100 | **1.5653** | 2.7669 | INT8&INT4&FLOAT |
| res34_cifar100 | 1.3383 | **1.3293** | INT8&FLOAT |
| res34_cifar100 | **1.3383** | 1.7104 | INT8&INT4&FLOAT |
| mobilenet_imagenet | 1.6358 | **1.6245** | INT8&FLOAT |
| mobilenet_imagenet | **1.6358** | 1.7313 | INT8&INT4&FLOAT |

Table 1: Experimental results

### 6.1.2 ALGORITHM SETTING

Before quantization process, we will process whole calibration dataset in full precision and find the min and max value in dataset for a layer's input and compute $\Delta$. We use this setting because we want to enlarge the noise and get a obvious experimental results. In CIFAR 10 VGG experiments, under this setting, we cannot find an appropriate layer to quantize because all $\Delta$s are large or $Choice$s variable are zero. Thus, we use the min/max on current quantization vector like HAWQ'sDong et al. (2019a) experiments to compute $\Delta$. To gain a high performance model as radically as possible, we set $error_{min} = 0, error_{max} = 0.1$. For CIFAR100 and ImageNet-100 experiments, we uses $\mu = 0.6$ in experiments, in MNIST and CIFAR10 dataset, the $\mu$ is 0.4. And the value of $\mu$ is adjusted during our experiments.

### 6.2 EXPERIMENTAL RESULTS

In this part, we also show the range of noise which is introduced by the different quantization level in the input of the layer. In our experiments, we find that INT8 quantization level brings less than 1e-2. Only few layers would be larger than this level (usually less than 5e-1), For INT4, the quantization noise is less than 5e-2 and few layers would larger than 1, and these layers whose noise larger than 1 should omit in quantization level.

In our algorithm, we substitute secant line for gradient in Eq. 5, which expands the applicability of Fig . 5 beyond the mathematical neighborhood concept into the 1e-2 ball. Actually we find that the gradient can used in Eq.5 only when the noise is less than 3e-4. However, secant line would fail when the noise is larger than 5e-2, which is smaller than the noise by INT4.

In INT8 level quantization, our algorithm can find a model with a lower loss function value than full-precision models. When the quantization noise is larger than 5e-2, secant line would fail in some cases.

## 7 CONCLUSION

This paper shows that quantization technology can improve the model's performance, i.e., gain a lower loss. Moreover, based on our analysis, we propose a Radical Mixed-Precision Inference Layout Scheme, which could produce a quantized model which is better than the full-precision model. We also show that residual networks are very resistant to noise. This means that the performance of a SOTA residual network is stable for any quantization algorithms.

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
