# OpenReview forum: "Mixed-Precision Inference Quantization: Radically Towards Faster inference speed,  Lower Storage requirement, and Lower Loss"
_ICLR.cc/2023/Conference — Submitted to ICLR 2023_

### Official Review · Reviewer_avng · 2022-10-21

**Confidence:** 5
**Clarity, Quality, Novelty And Reproducibility:** The paper is not very well written, a…
**Correctness:** 3
**Technical Novelty And Significance:** 1
**Empirical Novelty And Significance:** 1
**Recommendation:** 1

**Details Of Ethics Concerns:**

As shown above, I believe the analysis presented is almost identical to a previous paper I referred to.

**Strength And Weaknesses:**

The biggest concern I have with this paper is that, as far as I know, it is a serious case of using prior results without giving credit, nor adding any novel contribution on top. The analysis starting in eq. (4) has already been established in [1] who also used Chebyshev's inequality to obtain bounds on the accuracy in the presence of quantization similar to the way this paper arrived to eq. (8). Similarly, using this analysis to formulate a mixed-precision quantization strategy was proposed in [2].

[1] Sakr, Charbel, Yongjune Kim, and Naresh Shanbhag. "Analytical guarantees on numerical precision of deep neural networks." International Conference on Machine Learning. PMLR, 2017.
[2] Sakr, Charbel, and Naresh Shanbhag. "An analytical method to determine minimum per-layer precision of deep neural networks." 2018 IEEE International Conference on Acoustics, Speech and Signal Processing (ICASSP). IEEE, 2018.

**Summary Of The Paper:**

The paper tries to provide a method for mixed-precision quantization. A statistical analysis is provided to quantify the impact of quantization on model accuracy. This method is not novel.

**Summary Of The Review:**

As far as I am aware, this paper has used results from prior works without proper credit. On top of that, there is nothing really novel presented in the paper. Finally, there are almost zero empirical results, only a few sentences provided on a supposed accuracy using quantization.

---

### Official Review · Reviewer_JF51 · 2022-10-24

**Confidence:** 4
**Correctness:** 3
**Technical Novelty And Significance:** 2
**Empirical Novelty And Significance:** 2
**Recommendation:** 3

**Clarity, Quality, Novelty And Reproducibility:**

Clarify:
- Many statements are inaccurate or even wrong. For example, "No study examines how to increase the performance of a model using quantization technology" -> This is so confusing because the exact opposite is true for most quantization studies.

Quality:
- The paper would benefit from careful proofreading. In particular, the paper has many grammatical errors that severely hurt the work's readability. For example,
- "a mixed-precise quantization model"-> "a mixed-precision quantization model".
- "The loss function of the quantized model is usually higher than the full-precision model" -> It is unclear what "higher loss function" means. Probably "the accuracy loss of the quantized model is usually higher than the full-precision model".
- Almost all references are without parenthesis.

Novelty:
- The mathematical analysis of computational noise robustness seems to be novel although the reviewer has not dived deep into its detailed descriptions.

Reproducibility
- The paper provides a very limited description of its implementation and hyperparameter settings, making reproducing its results not very easy.

**Strength And Weaknesses:**

Strengths:
- The paper offers some interesting perspectives on why neural networks are robust to quantization from a computational noise perspective.
- The paper also offers some mathematical justification for why residual networks are robust to noise.

Weaknesses:
- There is no clear definition or description in terms of the problem this paper tries to address. Is the main goal to provide a mathematical justification on why DNNs are robust to quantization, or is the main goal to propose a better method for mixed-precision of quantization. From the reviewer's perspective, the first one is a very hard problem and may not even be true because, in practice, different models can have very different sensitivity to quantization (e.g., some generative models are very sensitive to quantization), and even the same model can have drastically different quantization results depending on how many bits are used for quantization and the quantization schemes. Unfortunately, none of those are covered in the theoretical analysis. If the goal is the latter one, then clearly the paper also fails to achieve that goal because the evaluation is largely inadequate due to the small datasets and models.
- Evaluation is very much done using toy datasets and small-scale models, making it hard to be convinced that the observations (e.g., lower loss from quantized models using the proposed method) can be generalized.
- There is a big improvement room for the writing quality of this paper.

**Summary Of The Paper:**

This paper introduces a method, called "Radical Mixed-Precision Inference Layout Scheme" to obtain a mixed-precision quantized model that has lower loss than the full-precision model. The paper provides mathematical justification on why deep neural networks are robust to quantization. Evaluation of CIFAR10 and several residual networks show that the proposed method can obtain quantized models with lower loss than the corresponding full-precision models.

**Summary Of The Review:**

Although offering some interesting perspectives on why DNNs are robust to quantization and the proposed mixed-precision quantization scheme is able to show some promising results on small datasets, the paper still has a large room for improvements in its writing quality and evaluation.

---

> ### Author Response · Authors · 2022-11-19
> **polished paper and add more experiments**
>
> Really thank you for your review and based on your review, we polish our paper, adjust our paper's organization and add more experiments to prove our work.
>
> Our main purpose of this paper is giving a method which can make the value of loss function of model lower.  The paper is organized as follow:
>
> *Start point: 1.the fluctuation of loss function value of a model is determined by the inner product of layer's input quantization error and the layer's gradient. 2. We can control the direction of quantization error to make the   inner product be negative which means quantized model has lower loss function values.
>
> *Mapping math concept into engineering and Algorithm: 1. The gradient only works in model's neighborhood, we should use secant line to expand the Eq.5 work filed. 2. The direction should be mapped into round operation, like floor (all quantization error is positive) or ceil function (all quantization error is negative). 3. Not all layer should be quantized for the ML's probabilty nature. And we give the a method to determine whether a layer should be quantized.
>
> *limitation of algorithm: In practice, algorithm does not work well in resnet 101. Thus we prove the limitation of algorithm by giving a stronger conclusion, i.e., theorem 1.
>
> *experiments: We use 4 dataset and 9 models in experiments
>
>
> Theorem 1  is an added results. To make theorem 1 more clearly, we show that only when the quantization noise is small enough (usually INT8 satisfies this requirement), the theroem 1 is correct.

---

### Official Review · Reviewer_kdep · 2022-11-03

**Confidence:** 3
**Correctness:** 3
**Technical Novelty And Significance:** 3
**Empirical Novelty And Significance:** 2
**Recommendation:** 5

**Clarity, Quality, Novelty And Reproducibility:**

Clarity: The paper is generally easy to follow, although I am not able to understand all the derivation.

Quality: The experimental results are not enough.

Novelty: The derivation looks interesting to me.

Reproducibility: Not so easy, but can be reproducible in the codes are provided.

**Strength And Weaknesses:**

Strength

The author tried to prove that for a specific full precision model, there always exists a quantized model which can reach or even outperform the accuracy of the full precision baseline.

Weaknesses

1. Some assumptions in the derivation are confusing for me. For example, in Eq. 7, why e and p are independent to each other? Also, above Section 4.1.2, the author said "this is why channel-wise quantization methods are booming", what is the relationship between "channel-wise" and equations 4-6?
2. The author said "Thus we have to choose the model which is far from the SOTA model". Is this statement related with Theorem 1, to say, for the (near) SOTA, your method cannot improve model performance too much? If so, I cannot understand the logic. Looks like you have to select some non-SOTA models as the baseline. Then even your quantization can be better than non-SOTA fully precision baseline, it might still be worse than some other quantization methods based on SOTA baseline?
3. The author said "like V100 GPU, only support INT8,INT16 and INT32 computing in hardware". If you did not intend to give the running time, you don't have to be restricted with the GPU runtime of some specific hardwares. Therefore, you can try more mixed precision such as lower bits (i.e. 2/4/6 bit).
4. It is better to compare with some other network quantization methods.

**Summary Of The Paper:**

This paper tried to explain a theoretical way to find a quantization model which is better than the anchor full precision model. My main concern is about the experimental results. There are only a few of results which are not persuasive. Moreover, there is no comparison with previous works.

**Summary Of The Review:**

Please check the Strength And Weaknesses. I think the current version is hard to be accepted since the results are not comprehensive. However, I am willing to increase the grade if the author can provide more evidence to persuade me.

---

### Decision · Program_Chairs · 2023-01-20

**Decision:**

Reject

**Justification For Why Not Higher Score:**

All reviewers favor reject.  The experimental section of the paper is far below standards, entirely lacking comparison to prior work.

**Justification For Why Not Lower Score:**

N/A

**Metareview: Summary, Strengths And Weaknesses:**

This paper presents a method for quantizing neural networks.  Reviewers unanimously favor rejecting the paper, citing numerous concerns including presentation, novelty and discussion of prior work, and lack of experimental validation.  The author response does not sufficiently resolve these concerns.  The only experimental results are that of the proposed method run on different base networks.  Network quantization is an active topic with significant prior work; there are no results on standard benchmarks, nor any attempt to compare to previously published methods.  This is far below the expected experimental standard in this subject area and is a clear indicator that the paper is not ready for publication.